# Resistance to High-Temperature Oxidation of Ti-Al-Nb Alloys

**DOI:** 10.3390/ma15062137

**Published:** 2022-03-14

**Authors:** Joanna Małecka

**Affiliations:** Faculty of Mechanical Engineering, Opole University of Technology, 45-271 Opole, Poland; j.malecka@po.edu.pl; Tel.: +48-77-4498466

**Keywords:** oxidation, intermetallic, Ti-Al alloys, high temperature

## Abstract

The research presented in this paper concerns the assessment of the resistance to high-temperature oxidation behaviour of a Ti-46Al-7Nb-0.7Cr-0.1Si-0.2Ni alloy and the explanation of the role of niobium during oxidation processes. The basic problem concerned the evaluation of the resistance of the studied alloy to cyclic oxidation in an air atmosphere, with particular attention to the influence of temperature, surface roughness and cooling rate from heating temperature to room temperature. The issue analysed was the effect of niobium addition on the corrosion kinetics as a high-melting element causing improved oxidation resistance, contributing to the reduction in the oxidation rate.

## 1. Introduction

The alloys whose structure consists of intermetallic phases of the Ti-Al system are attractive materials due to their properties which allow using them in many areas of activity, including commercially in aerospace, aircraft and automobile applications. These alloys, thanks to their properties, are used even at temperatures above 600 °C, where classical titanium alloys are not suitable for use due to low creep and oxidation resistance [1,2]. Recent research works [3,4,5,6,7,8,9,10,11] largely concern improving the oxidation resistance of Ti-Al intermetallic alloys.

This issue is currently the main problem causing application limitations of the discussed alloys, especially for components intended for long-term operation at cyclically varying temperatures. Increasing the resistance to oxidation is possible, among other things, by modifying the chemical composition with suitable alloying elements, which is the subject of this work. 

The oxidation of intermetallic Ti-Al alloys is significantly dependent on their chemical composition. A slowly growing, continuous and compact scale can exhibit protective capabilities, thus providing resistance to high-temperature oxidation of materials. Such properties are exhibited by alloys that are characterised by the ability to form an oxide layer based on oxides of Al_2_O_3_, among others [12]. The oxidation kinetics of Ti-Al intermetallic alloys depend mainly on the temperature, but also on the cooling rate [13], surface preparation [14], the degree of contamination and the type of microstructure [15], and the presence of nitrogen [16,17], water vapor [18,19] or exhaust gas [20]. Alloys containing more than 65% Al have the ability to form a favourable protective Al_2_O_3_ layer. At lower aluminium concentrations, the activity of this element is less than that of titanium, while TiO oxide is more stable thermodynamically. Practically, this oxide never forms, and TiO_2_ oxide is formed instead. Therefore, alloys based on Ti_3_Al and TiAl intermetallic phases in which the Al content is lower (42–48% Al) undergo faster oxidation. Therefore, these alloys should contain alloying additives that improve their oxidation resistance [21]. It is generally believed that small amounts of Nb significantly reduce the kinetics of oxidation in air, leading to the formation of a continuous layer of nitride at the oxide–metal interface that controls downstream oxidation, and similar results were observed during the oxidation of the Ti-47Al-2Cr-2Nb and Ti-45Al-2Cr-8Nb alloys [22]. Nb seems to be especially important because it is concentrated in the substrate near the phase boundary and limits formation of the solid solution α-Ti(Al,N,O). Nb, being a β-forming element in relation to titanium, promotes the formation of a poorly soluble phase β-Ti(Al,N,O) of nitrogen and oxygen. It limits the diffusion of interstitial elements to the core, and as a consequence, it slows down the increase in oxidation products.

This paper assumes that the increase in oxidation resistance of the discussed alloys can be achieved by the introduction of Nb as an alloying element, which can reduce the rate of oxidation of intermetallics. Optimal Nb content is in the range of 4–10 at.%, so this research was concentrated on an alloy containing 7% Nb. The main goal of this work was to determine the range of oxidation resistance of an alloy containing 7% Nb. On the basis of an analysis of corrosion products and diffusion processes in the metallic substrate and oxide layer, the mechanism of corrosion at high temperatures was determined. The next stage of the work consisted of determining the influence of increased niobium content on the course of oxidation, with the main focus on the influence of niobium on the rate of the process. The most common alloy studied in the literature for resistance to high-temperature oxidation is the alloy containing 2% Nb [22,23]. In this work, the literature data on the alloy Ti-48Al-2Cr-2Nb [24] were treated as a comparison and reference for the investigated Ti-46Al-7Nb-0.7Cr-0.1Si-0.2Ni alloy. This paper presents the results of reference alloys at temperatures below 875 °C only for informative results, not for comparison.

The most common oxidation process is isothermal oxidation; however, for practical reasons, the introduction of cyclic temperature changes is a process designed to approximate the test conditions to the actual working conditions of the material. Components of structural equipment may work not only in high-temperature oxidation conditions but also in cyclic operation conditions, requiring cyclic start-up and shut-down. During cyclic oxidation, the scale is usually detached from the substrate. This is the primary reason for the poorer resistance of the material under these conditions. The main factors of cyclic oxidation determining the oxidation rate and the thickness of the oxide layer formed are heating temperature and cycle frequency. It is worth noting that the compactness of the oxidation product layer formed on multicomponent Ti-Al alloys is also determined by other factors important for methodological reasons. Apart from the chemical composition of the alloys, the atmosphere and the temperature at which they operate, the cooling rate and the surface condition of the alloys prior to operation may play important roles [2,13]. Therefore, the present study considered not only the possibility of improving the oxidation resistance by the effect of Nb addition, but also the sensitivity of the alloys to particular factors of cyclic oxidation.

## 2. Material and Test Methods

A two-phase multi-component Ti-46Al-7Nb-0.7Cr-0.1Si-0.2Ni alloy was selected as the test material. The material in the form of a cylinder with a diameter of about 69 mm was purchased from Flowserve Corporation Titanium and Reactive Metals Foundry (RMI Titanium Company Inc., Pittsburgh, PA, USA). This alloy was prepared by the arc melting process. In order to remove the internal defects and obtain a homogenised sample, the castings were processed by the HIP (hot isostatic pressing) method. The surfaces were abraded on abrasive paper of 800 grade or by abrasive blasting using alundum of F40 grain and finally cleaned with acetone. Specimens with dimensions of ≈20 mm × 15 mm × 2 mm and surface roughness Ra = 0.06 μm (after grinding) and Ra = 5.8 μm (after abrasive blasting) were prepared for testing. The surface roughness was measured using a TOPO-01P profilometer (Digital Surf Companies, Besançon, France) with a diamond stylus radius of 2 µm.

Cyclic oxidation experiments were realised at temperatures of 875 °C, 900 °C, 925 °C, 950 °C and 975 °C for 60 cycles in atmospheric air. The samples were heated up in the desired atmosphere with the tube furnace (high-accuracy temperature sensors: ±1 °C) (Czylok, Jastrzebie-Zdroj, Poland), then annealed in a given temperature, and subsequently cooled down to the room temperature. Each traction cycle consisted of heating the samples to a specific temperature, annealing at this temperature for 1 h with subsequent cooling, and holding at room temperature for 15 min. The gravimetric method was selected to measure the reaction speed. The control of weight change was performed on a RADWAG precision scale with accuracy of 10^−4^ g. Trials were repeated three times and the presented test results were averaged. During cyclic oxidation in air, in addition to the temperature, the variable factors were the surface condition as determined by the Ra and the value of the cooling rate from the heating temperature to room temperature (18 °C). The microstructure was then observed using a HITACHI S-4200 scanning electron microscope (Hitachi High-Tech in Europe GmbH, Krefeld, Germany) operating at 20 kV. Secondary electron emission (SE) and backscattered electron emission (BSE) were used in the tests. Microanalyses of chemical composition were performed with the EDS method using the Therm NORAN attachment with System Six coupled with HITACHI microscope. Phase composition tests were performed by X-ray diffraction using the JDX-7S diffractometer (JEOL Companies, Tokyo, Japan) from JEOL using a copper X-ray tube (λCuKα  =  1.5406 Å). The recording was carried out by a step method with a step of 0.05° and a count of 3 s in the range of 20° to 90° 2θ. Phase identification was carried out using the ICDD database.

## 3. Test Results

### 3.1. Microstructure Analysis

The structure of the tested Ti-46Al-7Nb-0.7Cr-0.1Si-0.2Ni alloy and the XRD microanalysis is shown in Figure 1. According to the microstructure classification of Ti-Al alloys, the tested alloy can be described as duplex. It reveals the clear contrasts between the phases. The TiAl and Ti_3_Al phase form a plate system and the TiAl phase is present locally in the granular form. Moreover, an irregular concentration of Ti_3_Al phase is found on the border of the plate grain system. The distance between Ti_3_Al plates in a plate mixture Ti_3_Al + TiAl is from ca. 0.5 μm to ca. 1.5 μm. The thickness of TiAl phase plates is 4 to 10 times higher than Ti_3_Al phase.

### 3.2. Determination of Kinetics of Cyclic Oxidation

Oxidation at lower temperatures (from 875 °C to 925 °C) over the range of the 60 performed cycles only results in mass gain (Figure 2). Oxidation at 950 °C and 975 °C induces mass gain only in the first cycles. For 950 °C, this occurs after about 30 oxidation cycles and for 975 °C, after 14 oxidation cycles. At 975 °C, after 30 oxidation cycles, the sample reaches a mass close to the initial one, and with further cycles, only mass loss takes place.

For the Ti-48Al-2Cr-2Nb alloy treated as a reference alloy in this work, the course of cyclic oxidation in the temperature range of 800 °C to 875 °C is shown in Figure 3. The temperatures of 800 °C, 825 °C and 850 °C showed relatively small mass gains throughout the test. At 875 °C, the reference alloy achieved a mass gain of 1.2 mg/cm^2^ after only about 30 oxidation cycles (Figure 3), while the tested alloy with 7 at.% Nb content had a mass gain of about 0.8 mg/cm^2^ and about 1 mg/cm^2^ after 60 cycles at this temperature (Figure 2). At 900 °C and 925 °C, the products formed of the Ti-46Al-7Nb-0.7Cr-0.1Si-0.2Ni alloy showed suitable adhesion to the substrate throughout the test. On the other hand, for the reference alloy at 900 °C and 925 °C, weight loss can be observed as early as 10 and four oxidation cycles, respectively (Figure 3). The results show that the alloy with 7% niobium content is characterised by higher resistance to high-temperature oxidation than the alloy containing 2% niobium, which manifests itself in a reduction in the oxidation rate.

Studies on the effect of high cooling rates (water) and very low cooling rates (1 °C/min) are presented in [2] and [13], respectively. Although at low cooling rates, less spalling of the scale would be expected due to its deformation by creep at high temperature, the experimental results indicate that spalling increases. The reason for this is not clear, although it is speculated that the micro-crack formed at high cooling rates, oriented perpendicular to the alloy surface, increases the resistance to spalling as it decreases the stresses in the oxide layer. In contrast, low cooling rates facilitate lateral growth strain [25], which increases the stresses.

The study of the effect of cooling rate was carried out for Ti-46Al-7Nb-0.7Cr-0.1Si-0.2Ni alloy cyclically oxidised at 900 °C and 925 °C (Figure 4). For cycles of heating at 900 °C with subsequent cooling in compressed air, higher mass gains were found than after cooling in still air. For the temperature of 925 °C, differences between cooling methods are practically non-existent.

The higher surface roughness (Ra = 5.8 μm) contributes to a slight increase in mass gain for the temperature of 900 °C (Figure 5). This is due to the specific surface area, which is not included in the macroscopically determined total surface area. However, at higher temperatures, despite this specific surface area, the situation changes.

At 925 °C, larger mass gains are observed for samples with low surface roughness, and at 975 °C, from “smooth” surfaces, the scale easily chips off from the metallic substrate, while it adheres well to surfaces with high roughness.

### 3.3. Effect of Cyclic Oxidation on the Microstructure and Chemical Composition of the Scale

The diffusion processes become more active with increasing temperature and oxidation time. This was observed at 950 °C and 975 °C (Figure 2). As indicated on the basis of the performed investigations, analysing the images of the surfaces and cross-sections, the formed layer is always characterised by the order of sublayers. An increase in process temperature, however, results in greater thicknesses of the oxide layer formed. On the oxidised alloy, there are columnar crystallites (Figure 6). At the lower temperatures (875–925 °C), they are very small, and therefore, observations were made at magnifications of 4000×. The oxide layers formed, which adhere well to the metallic substrate throughout the test in this temperature range, undergo spalling under dynamic bending loads already after oxidation at 875 °C (Figure 7).

The cross-sections and metallographic specimens of the scale also show that the sublayers do not fully cover the entire surface (Figure 8). EDS analysis performed in location 1 according to Figure 8 (a clear dominance of titanium can be seen) reveals that the outer sublayer is rutile (Figure 9a). Locally, another sublayer is in direct contact with the oxidant. Between the rutile and the next sublayer, there are nano-voids (in BSE as black). Immediately below the columnar rutile crystallites is the next sublayer, which contrasts heterogeneously grey in the BSE. Microanalysis of this sublayer (Figure 9b) shows that Al dominates there, with a much smaller participation of Ti. Thus, it can be concluded that there is a predominance of aluminium oxides in this layer. The third oxide sublayer extends up to the metallic substrate and it has the greatest thickness. 

Ti oxides are dominant in this sublayer but Al oxides are present, too (Figure 9c). The effect of the oxidation of the Ti–46Al–7Nb–0.7Cr–0.1Si–0.2Ni alloy carried out in the temperature range 875–925 °C is not only the formation of reaction products but also the course of diffusion processes in the metallic substrate. There is a bright band of discontinuous groupings of elements with large atomic numbers (Figure 8b, point #4). This is an area rich in elements with high atomic numbers such as Nb, Ni and Cr (Figure 9d). Such observations regarding the structure of oxidation products were confirmed by quantitative microanalysis studies (results are summarised in Table 1).

Investigations of the tested alloys with surface prepared by grinding after oxidation at 950 °C and 975 °C were impossible to carry out without violating the compactness and cohesion of the layer. Already after about 30 h of oxidation at 950 °C and after 14 h of oxidation at 975 °C, the products only locally retained their continuity. The observation of the external surface of the products indicates that the formed columnar rutile crystallites are much bigger in size than after the oxidation at 875 °C to 925 °C (Figure 6 and Figure 10). The formation of such products, as shown in Figure 10, requires explanation. These crystallites are formed not only by out-core diffusion of titanium but also by surface diffusion. The interpretation can be based on the model presented by Jungling and Rapp [27]. Studies by Rapp and co-workers [28,29] using hot-stage environmental SEM and video camera recordings have shown that at lower temperatures, surface diffusion is the fastest process that leads to whisker formation. A slight share of lattice diffusion causes their extension, i.e., growth in the perpendicular direction. As temperature increases, the participation of lattice diffusion becomes significant, so oxide growth occurs rapidly over the entire grain surface, while in the vicinity of the dislocation, surface stresses and energy are minimised by cavity growth in depth. The observed crystallites therefore form with the interaction of both types of diffusion. This proves that there is an out-core transfer of titanium ions through the product layer of both Al_2_O_3_-rich and TiO_2_-rich products. At the same time, it proves that a sublayer of Al_2_O_3_ sufficiently homogeneous to be capable of inhibiting the diffusion processes and, therefore, of markedly increasing the resistance to oxidation, was not formed. The outer surfaces of samples oxidised at 950 °C and 975 °C for 14 and 30 h, when cooled to ambient temperature, show fragmentary spalling of the essential part of the oxidation products. As the holding time at room temperature increased, stresses accumulated and further “delayed” spalling occurred until the essential part of the layer was completely lost (Figure 11 and Figure 12).

The surface of the oxidised alloy after spallation of the basic layer of oxidation products is shown in Figure 13. The structure of these products is extremely fine-grained. The process of such fragmentary spalling occurs not only in one cycle but also in many previous cycles. In the micro-areas in Figure 12, differences in composition become apparent. Thus, the surface fragments in which point A was chosen according to Figure 12 are clearly enriched in titanium (about 76.7 at.%) and contain aluminium (Figure 14a, Table 2), while the composition of the sublayer that remained after the spallation, in which points B and C were chosen, besides titanium and aluminium, also contains nickel, chromium and niobium (Figure 14b,c). This structure of the part of the sublayer remaining on the metallic substrate after the basic layer of oxidation products has spalled off indicates that they are multiphase and multilayer but compact. The morphology and layering are no different from those obtained at lower temperatures (Figure 15). However, it can be observed that micropores develop more actively. This is related to the increase in temperature.

The banded contrast between the product layer and the metallic substrate is an incentive to check whether the chemical composition of the different sublayers was affected by the temperature increase. To this end, microanalyses were performed at multiple points in the layer (Figure 16, Table 3).

Analysis of columnar crystals (Figure 16a) indicates that these are rutile crystals. The next dark-contrasting oxide sublayer contains the most Al, which may suggest that Al_2_O_3_ is the dominant phase (Figure 16b). However, the presence of titanium (which indicates that it is bonded in TiO_2_) manifested, among others, by bright, very fine nanoparticles means that Al_2_O_3_ is not the only phase component of this sublayer. The third sublayer extending down to the metallic substrate contrasts much brighter, although locally, the contrast is deeper. Microanalysis in point 3 (according to Figure 15) selected in this sublayer may indicate the presence of titanium oxide, aluminium and alloying element oxides (Figure 16c). Similarly, a bright band enriched with Nb, Cr and Ni was observed at a lower temperature (Figure 16d). Microanalysis of the chemical composition in different areas of the metallic substrate is shown in Figure 16e,f.

As in the study of the structure and composition of the products obtained when oxidation was carried out at 875 °C to 925 °C, the cross-sections of the scale, the formation of which did not introduce additional interactions on the layer, were observed and analysed under the studied conditions (975 °C). The cross-sectional areas of the oxide scale formed on the alloy after grinding show a thin, uniform layer (due to chipping during cooling) of about 6 μm thickness, with a running bright band enriched in Nb (Figure 17). Stress build-up in the scale leads to the initiation of microcracks, which promotes the formation of a multilayer scale. The extension of processing time causes the formation of successive sublayers, which is shown by the photos taken during microanalyses of cross-sections of the oxide layers formed. As the oxidation time increases here, the contact between the scale and the metallic substrate is locally broken and the construction of the second layer begins. The sites of fragmentary spalling of the scale after oxidation at 975 °C are shown in Figure 18 and Figure 19.

The way of surface preparation, especially higher roughness, definitely has a beneficial effect on the kinetics of cyclic oxidation, causing an increase in the cohesion of the oxide with the metallic substrate. This is because the diffusion processes occurring at the flat product–oxidation phase interface cause coalescence of vacancies, formation of micro-voids and their merging into bands (Figure 19). The structure of the product layer on the surface with Ra = 5.8 μm roughness is different. This is because the layer has an uneven thickness, and the diffusion processes in the direction of decreasing concentrations do not produce voids (Figure 20a). The voids do not form probably because the outer surface of the oxide is not parallel to the surface of the phase interface, and on the microscopic scale, the shear stress at each point of the phase boundary lies in a different plane, which does not promote stress accumulation. However, it should be mentioned that the chemical composition of the oxide sublayers formed does not change. A cross-section of the resulting products of the alloy whose surface was prepared using abrasive blasting is shown in Figure 20b. There are visible irregularities produced by the sample surface preparation by blasting. On the surface of the alloy prepared by abrasive blasting, no vertical columnar rutile TiO_2_ crystallites are formed. This is due to lower stresses, which prevent the rutile from growing vertically. The lower stresses present here result in less rutile growth compared to the stresses present on the ground surface.

XRD phase analysis results for Ti–46Al–7Nb–0.7Cr–0.1Si–0.2Ni after oxidation at 875 °C, 900 °C and 950 °C are presented in Figure 21, Figure 22 and Figure 23. The nucleating phase components are rutile TiO_2_ and Al_2_O_3_. Apart from these, there are also nitrides TiN, Ti_2_AlN and a metallic substrate phase (γ-TiAl phase). To obtain information about the phase at the metallic substrate, the scale was removed from the samples (Figure 21). 

## 4. Discussion

In order to determine the range of resistance of the Ti-46Al-7Nb-0.7Cr-0.1Si-0.2Ni alloy to oxidation in air, tests were carried out on cyclic oxidation of the alloy in the temperature range of 875 °C to 975 °C. It was found that with traditional surface preparation by grinding to a low Ra value (Ra ≈ 0.06 μm in the study), visible scale spalling occurs at 950 °C and above. This is because the diffusion processes occurring at the flat product–oxidation phase interface lead to the formation of microcavities and merging into bands. An increase in surface roughness up to Ra ≈ 6 μm increases the mass gain during oxidation at 900 °C (this is due to a larger surface development, which is not included in the macroscopically determined total surface area of the oxidised sample), while at higher temperatures (925 °C, 975 °C), it strongly reduces the processes of scale spalling. This is because higher surface roughness increases the cohesion of the metallic substrate with the oxide and causes a reduction in compressive stresses in the layer. This is because the layer has a non-uniform thickness and the diffusion processes that take place do not produce voids. Studies on the effect of high cooling rates (water) and very low cooling rates (1 °C/min) are presented in [2] and [13], respectively. At low cooling rates, less spalling of the scale would be expected due to its deformation by creep at high temperatures, but the experimental results indicate that spalling increases. The reason for this is not clear, although the micro-crack formed at high cooling rates oriented perpendicular to the alloy surface may increase the resistance to spalling. On the other hand, low cooling rates facilitate lateral growth strain, which increases compressive stresses in the oxide layer [25]. The study of the effect of the cooling rate on the oxidation of the Ti-46Al-7Nb-0.7Cr-0.1Si-0.2Ni alloy proved that increasing the cooling rate does not affect the acceleration or deceleration of the oxidation rate. This is confirmed by the observation of Yoshihara and Kim [2], as two of the six alloys they tested (oxidised cyclically at 875 °C), namely, the Ti-45Al-7.5Nb-0.3B-0.1Y alloy plus 0.2% each (W, Hf, Zr, C and O) and the Ti-47Al-2Cr-0.8Nb-0.2O alloy, were not sensitive to changes in the cooling rate. However, it should be noted that in their work [2], 200 oxidation cycles were performed at the analysed temperature, while only 60 cycles were performed in the present work.

The next stage of the research was the analysis of corrosion products and diffusion processes occurring in the oxide layer and in the metallic substrate. The analysis of the obtained results allows us to conclude that the oxidation of the tested alloy results in the formation of scale and transformations in the metallic substrate. They are caused by the out-core diffusion of alloying elements and the formation of phases and solid solutions as a result of the in-core diffusion of nitrogen and oxygen. Based on the investigations performed, the most general conclusion is that the oxidation products consist of three sublayers, and a Nb-rich band is formed in the metallic substrate (Figure 24 and Figure 25).

During the oxidation of the investigated alloy, it is not possible for a continuous protective layer of Al_2_O_3_ to form because the outer oxide layer always consists of TiO_2_. The reason for this is the main alloying element, titanium, which does not provide an opportunity for a protective layer to form on the basis of aluminium oxides. A characteristic feature of titanium is that it can have both cation and anion sublattice defected, depending on the oxygen partial pressure.

The formation of the outer rutile layer causes a change in the balance of the Ti and Al quantitative fraction and, as a result of the slower diffusion of this element, a mixed Al_2_O_3_-rich sublayer is formed under the TiO_2_ layer. The out-core diffusing titanium cations mostly migrate towards the surface. The resulting Al_2_O_3_ layer is a heterogeneous layer and it is not compact. It also includes, although in smaller amounts, rutile TiO_2_. The presence of rutile determines the further course of the oxidation process. Rutile has a defective anionic sublattice where the oxygen in-core diffusion takes place. Oxygen diffuses toward the product–metallic substrate interface, causing an increase in the thickness of the oxide layer due to the formation of titanium and aluminium oxides in this layer. The occurrence of defects in the rutile depends on the partial pressure of oxygen. At higher temperatures, the in-core diffusion of oxygen (through the doubly and singly ionised oxygen vacancies of the rutile) and the out-core diffusion of the titanium, aluminium and alloying elements occur. This produces a third mixed sublayer extending from the Al_2_O_3_-rich sublayer to the metallic substrate.

In the region of the metallic substrate of the investigated alloy, diffusion processes also take place in both oxidising atmospheres used and, therefore, a brightly contrasting, interrupted, niobium-rich micro-band forms in the metallic substrate in BSE. Niobium does not diffuse in-core simultaneously with Al and Ti, but remains in the metallic substrate, which relatively increases its concentration. Unfortunately, it was not possible to determine by X-ray methods in which compounds this element occurs in this band. A similar problem was observed in other works. Kim and Yoshihara [2] and Roy [31] did not identify Nb-containing oxides in the resulting products, despite the quite diverse content of this element in the alloys studied (2–15 at.%).

Through the product layer, nitrogen and oxygen diffuse into the metallic substrate. Oxygen is the main component of the formed products, while nitrogen does not form nitrides in it. It migrates toward the metallic substrate and combines mainly in Ti_2_AlN and TiN, and dissolves together with oxygen in the αTi(Al) solid solution. The possibility of forming such solid solutions is due to the equilibrium systems of Ti-O and Ti-N [32].

The influence of Nb on increasing the oxidation resistance of the investigated alloy is undeniable. Performed studies on the oxidation of Ti-46Al-7Nb alloys in relation to literature data [24] on the oxidation of Ti-48Al-2Cr-2Nb alloys show that the alloy with 7 at.% Nb content is characterised by higher resistance to high-temperature oxidation than the alloy containing 2 at.% Nb, which is manifested by a reduction in the oxidation rate and smaller mass gains during oxidation tests. To clarify, the effects of niobium on the structure and layer of oxidation products were analysed. In the analysed alloy, the products include niobium and other elements of the metallic substrate diversely distributed in the different sublayers. In the columnar sublayer of rutile crystallites, there are no elements such as Nb, Cr or Ni. In the dark band of the Al_2_O_3_-rich sublayer, Nb was observed in a much lower amount than in the next TiO_2_-rich sublayer. In the sublayer with balanced amounts of TiO_2_ and Al_2_O_3_, the Nb content is comparable to its local concentration in the metallic substrate. A definite increase in niobium content and occurrence of Cr and Ni was observed in the bright band, which is formed during oxidation, independently of the process temperature, oxidising atmosphere and other factors studied in this work (surface roughness, cooling rate). It was found that such a distribution of niobium in a narrow micro-band inhibits the diffusion processes through the sublayer closest to the metallic substrate. Why does this happen? Both allotropic varieties of titanium form terminal solid solutions with O and N [32]. The solubility of oxygen in α-Ti is about 34%, while that of nitrogen is about 23 at.% On the other hand, about 4% of oxygen and 2 at.% nitrogen can be dissolved in the β-phase. With the increase in aluminium content in these alloys, α_2_-TiAl and γ-TiAl phases are formed. In the γ-TiAl phase, oxygen dissolves in a very small amount, while nitrogen does not dissolve at all. The α_2_-TiAl phase is characterised by much higher solubility of both elements. The bidirectional diffusion that occurs during the oxidation of these alloys makes rutile present both at the product–metal interface and at the product–oxidant interface. Due to the high chemical activity of aluminium, in nanometre-sized regions, quite a significant part of Al will undergo selective oxidation and there will be a decrease in Al concentration in the metallic part of the alloy. This leads to a local disappearance of the γ-TiAl phase and an increase in the proportion of the α_2_-TiAl phase, and the substrate is enriched in nitrogen and oxygen. Al, O and N are alpha-forming elements and exhibit favourable solubility in α-TiAl. Nb is a beta-forming element with respect to titanium (it extends the occurrence of the β-Ti phase) and therefore, by forming the β-Ti phase, it limits the solubility of nitrogen and oxygen [33,34,35,36]. Nb in the alloy, concentrated in the substrate near the phase interface, limits the possibility of α-Ti(Al,N,O) solid solution formation and, as a beta-forming element relative to titanium, promotes the formation of a β-Ti(Al,N,O) phase with low O and N solubility. This limits the process of in-core interstitial element diffusion, which slows down the growth of oxidation layers. This is evidenced by the smaller mass increase in the tested alloy with respect to the Ti-48Al-2Cr-2Nb alloy during oxidation. 

In addition to the mentioned beta-forming role of Nb in alloys and the associated lower solubility of oxygen and nitrogen [33,34,35,36], the effect of niobium on increasing heat resistance is also related to other factors [35,37,38,39]:-Niobium inhibits film growth kinetics by decreasing titanium activity;-By reducing the activity of titanium, it increases the activity of aluminium, and supresses the growth of rutile TiO_2_;-Nb ions replace Ti ions, thus leading to a reduction in oxygen vacancies that reduce oxygen diffusion; and-Niobium hinders the mass transfer of TiO_2_.

As shown in this work, better corrosion resistance is provided by higher surface roughness and increased Nb content in the material, which, in the range of 7 at.%, reduces the oxidation rate compared to the reference alloy containing 2 at.% Nb. Taking into account the presented test results, it can be concluded that the addition of 7 at.% Nb causes changes in the structure of the metallic substrate and contributes to a significant improvement in resistance to high-temperature oxidation.

## 5. Conclusions

The addition of 7 at.% Nb causes a decrease in the oxidation rate, which is manifested by a 30% lower mass gain and better adhesion of the product layer compared to an alloy containing 2 at.% Nb.Multiphase scale is formed, which is always characterised by a sequence of sublayers: -Outer rutile sublayer;-Sublayer rich in Al_2_O_3_ and containing TiO_2_ in smaller amounts; and-Sublayer containing Al_2_O_3_, TiO_2_, and oxides of Nb, Cr and Ni.

In the oxidation tests, changes in the metallic substrate were observed, consisting of Al diffusion and the dominance at the oxidation temperature of the β-Ti(Nb,Al,N,O) phase with low amounts of dissolved oxygen and nitrogen.Nb forming a discontinuous band in the metallic substrate inhibits the diffusion processes taking place.The increase in surface roughness contributes to a decrease in the oxidation rate and inhibits oxidation product spallation processes due to a decrease in compressive stress and an increase in cohesion between the oxide and the metallic substrate.

## Figures and Tables

**Figure 1 materials-15-02137-f001:**
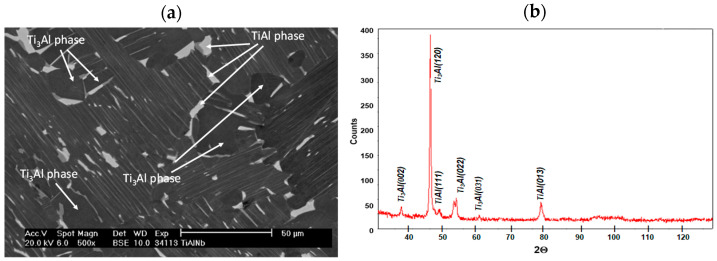
The (**a**) structure and (**b**) XRD microanalysis results of the Ti–46Al–7Nb–0.7Cr–0.1Si–0.2Ni alloy.

**Figure 2 materials-15-02137-f002:**
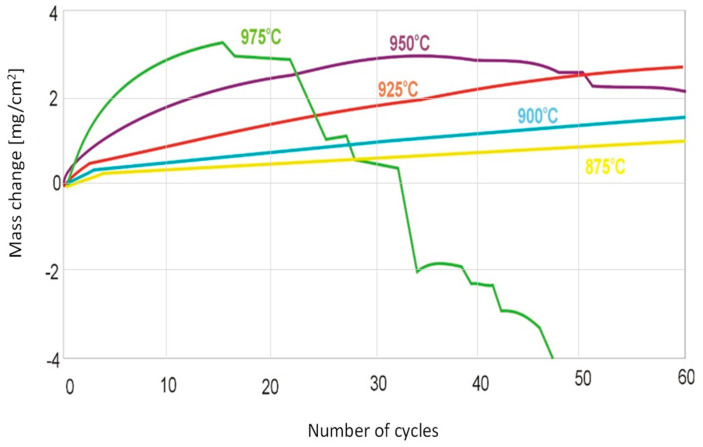
Mass change of Ti-46Al-7Nb-0.7Cr-0.1Si-0.2Ni per unit area with respect to exposed temperature.

**Figure 3 materials-15-02137-f003:**
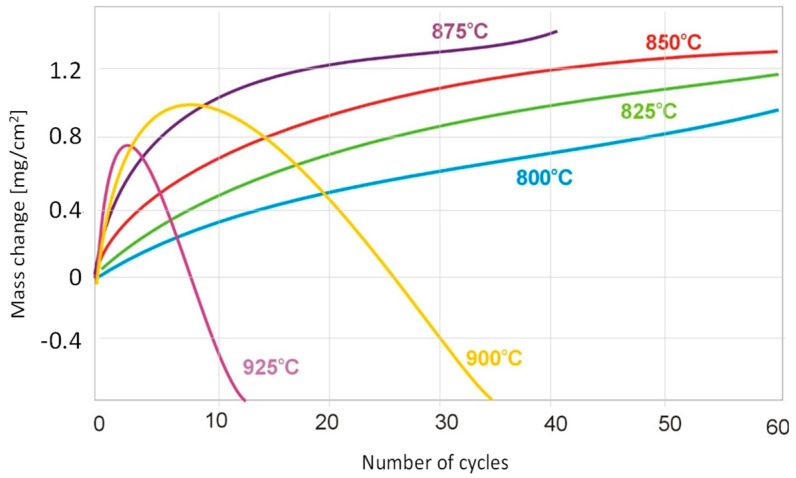
Mass change of Ti-48Al-2Cr-2Nb per unit area with respect to exposed temperature.

**Figure 4 materials-15-02137-f004:**
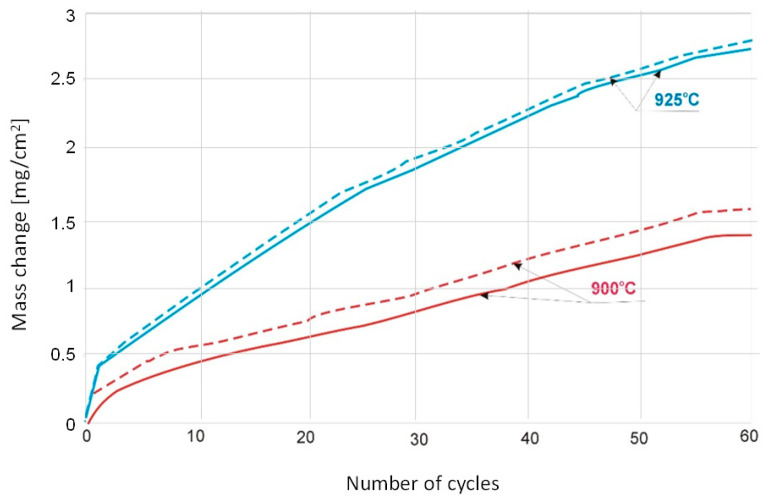
Mass change of Ti-46Al-7Nb-0.7Cr-0.1Si-0.2Ni per unit area with respect to exposed temperature (continuous curves—cooling in still air, dot curves—cooling in compressed air).

**Figure 5 materials-15-02137-f005:**
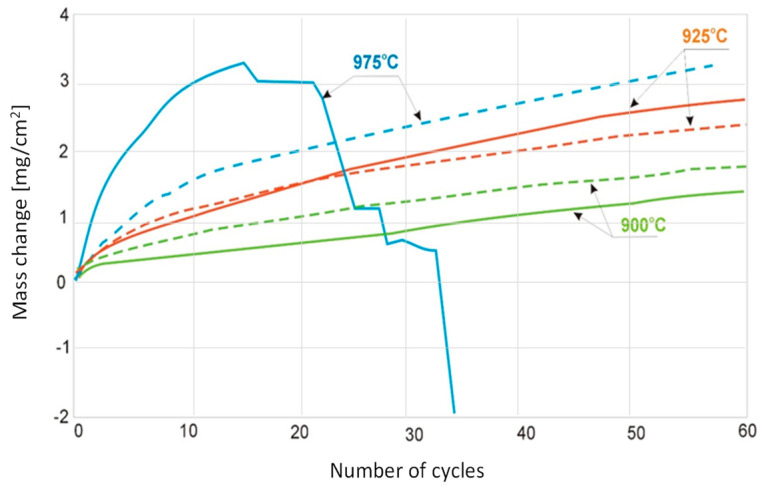
Mass change of Ti-46Al-7Nb-0.7Cr-0.1Si-0.2Ni per unit area with respect to exposed temperature (two surfaces: Ra = 0.06 µm—continuous curves and Ra = 5.8 µm—dot curves).

**Figure 6 materials-15-02137-f006:**
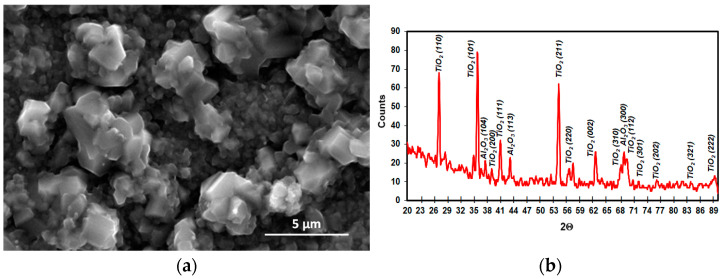
Surface of Ti-46Al-7Nb-0.7Cr-0.1Si-0.2Ni alloy exposed to air at 875 °C: (**a**) SEM micrograph—surface prepared by grinding [26]; (**b**) XRD microanalysis result—small columnar crystallites of TiO_2_.

**Figure 7 materials-15-02137-f007:**
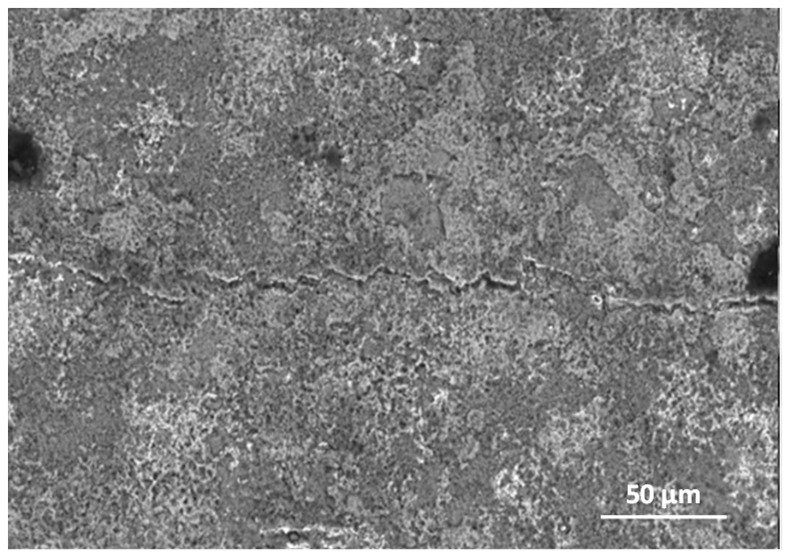
SEM micrograph of Ti-46Al-7Nb-0.7Cr-0.1Si-0.2Ni alloy exposed to air at 875 °C—surface prepared by grinding (fragmentary chipping of products).

**Figure 8 materials-15-02137-f008:**
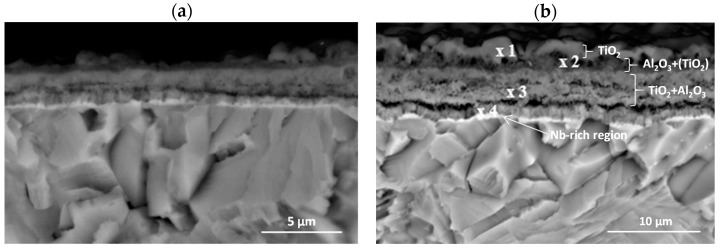
SEM-BSE images of Ti–46Al–7Nb–0.7Cr–0.1Si–0.2Ni alloy after cyclic oxidation at (**a**) 875 °C [26]; (**b**) 925 °C (surface prepared by grinding).

**Figure 9 materials-15-02137-f009:**
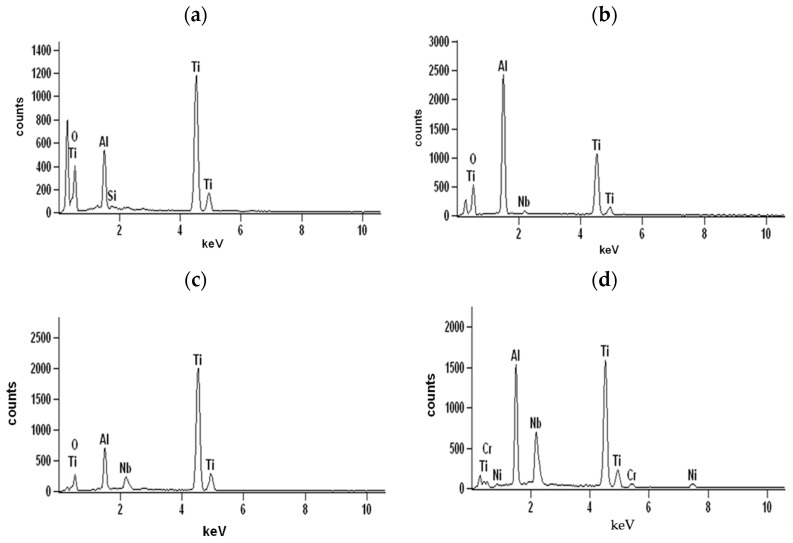
EDS analysis: (**a**) in point 1, (**b**) in point 2, (**c**) in point 3 and (**d**) in point 4 marked in Figure 8b.

**Figure 10 materials-15-02137-f010:**
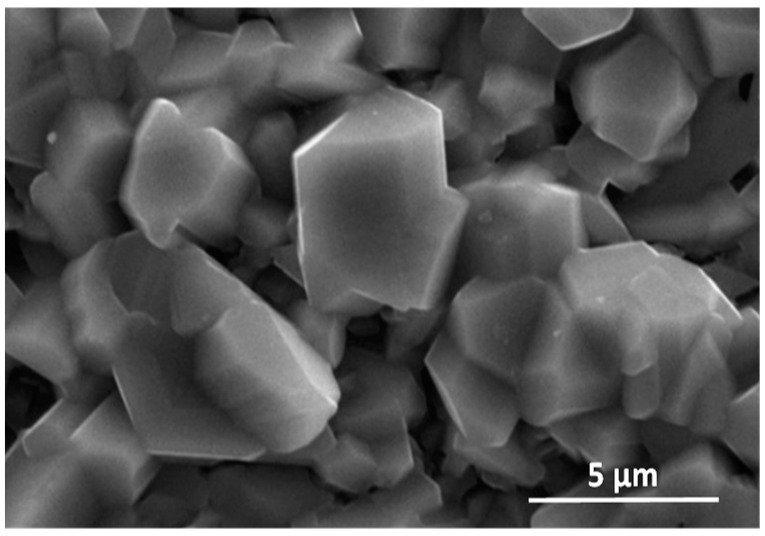
SEM micrograph of Ti-46Al-7Nb-0.7Cr-0.1Si-0.2Ni alloy exposed to air at 950 °C—surface prepared by grinding (bigger columnar crystallites of TiO_2_).

**Figure 11 materials-15-02137-f011:**
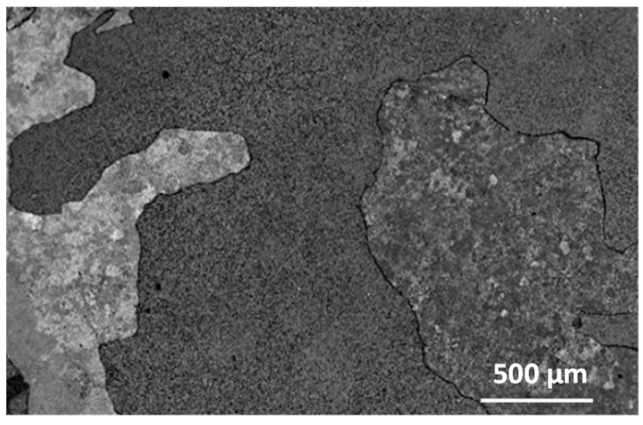
SEM micrograph of Ti-46Al-7Nb-0.7Cr-0.1Si-0.2Ni alloy exposed to air at 900 °C—surface prepared by grinding (fragmentary chipping of products) [30].

**Figure 12 materials-15-02137-f012:**
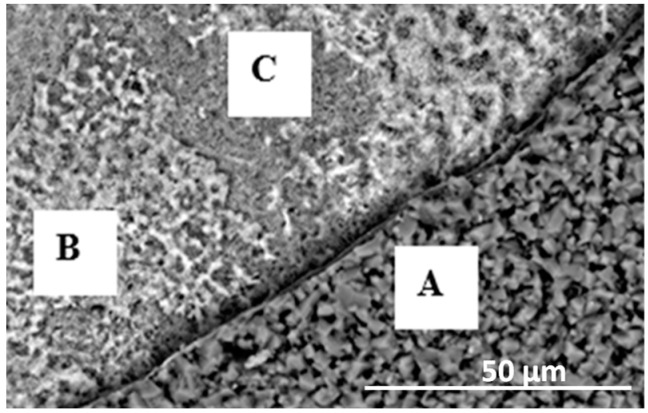
Surface of Ti-46Al-7Nb-0.7Cr-0.1Si-0.2Ni alloy after spallation of the basic layer of oxidation products (prepared by grinding). The structures of products in fields A, B and C are shown in Figure 13.

**Figure 13 materials-15-02137-f013:**
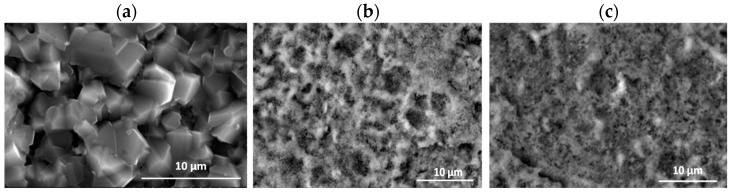
Fragment of the surface in fields (**a**) A, (**b**) B and (**c**) C marked in Figure 12.

**Figure 14 materials-15-02137-f014:**
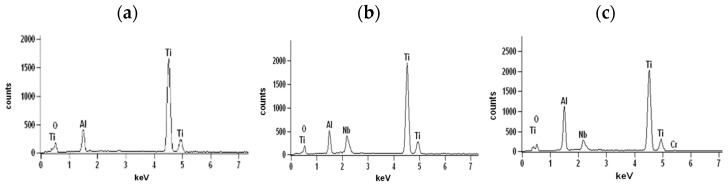
EDS analysis: (**a**) in area 1, (**b**) in area 2 and (**c**) in area 3 marked in Figure 12.

**Figure 15 materials-15-02137-f015:**
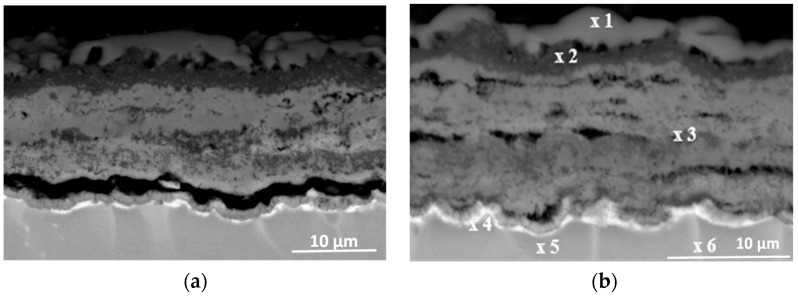
SEM-BSE images of Ti–46Al–7Nb–0.7Cr–0.1Si–0.2Ni alloy after cyclic oxidation at (**a**) 925 °C; (**b**) 950 °C (surface prepared by grinding) [26].

**Figure 16 materials-15-02137-f016:**
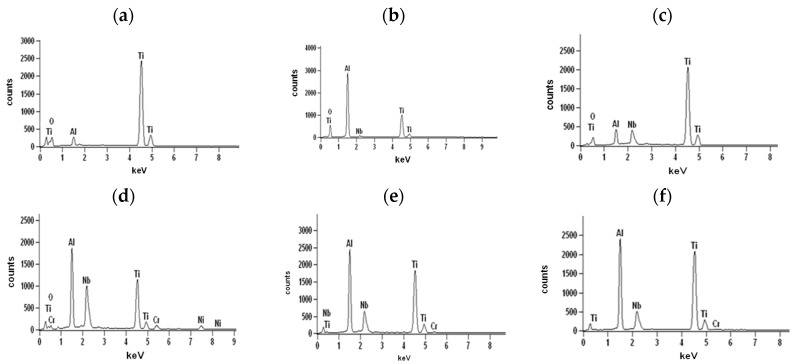
EDS analysis: (**a**) in point 1, (**b**) in point 2, (**c**) in point 3, (**d**) in point 4, (**e**) in point 5 and (**f**) in point 6 marked in Figure 15.

**Figure 17 materials-15-02137-f017:**
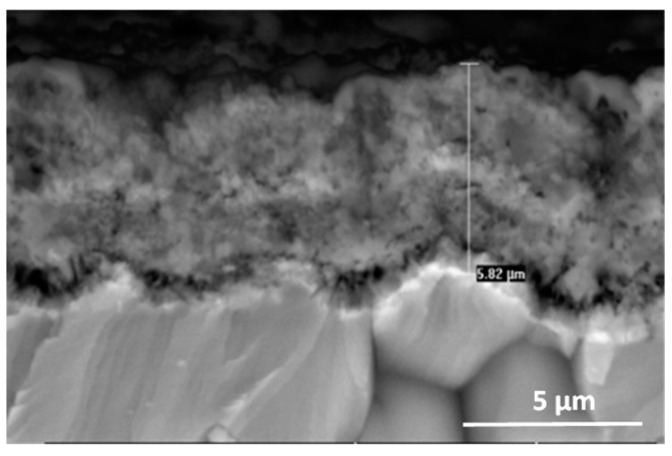
SEM-BSE images of Ti–46Al–7Nb–0.7Cr–0.1Si–0.2Ni alloy after cyclic oxidation at 975 °C (surface prepared by grinding).

**Figure 18 materials-15-02137-f018:**
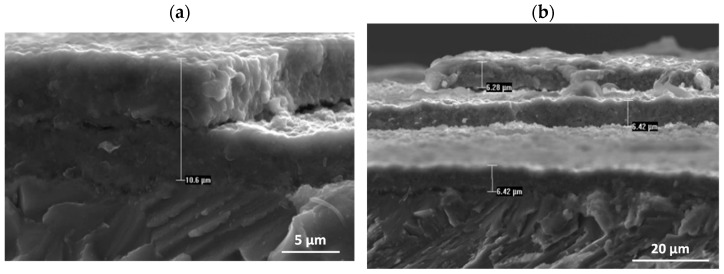
The cross-sectional areas of the scale formed on the alloy oxidised at 975 °C (surface prepared by grinding): (**a**) high-magnification, (**b**) low-magnification.

**Figure 19 materials-15-02137-f019:**
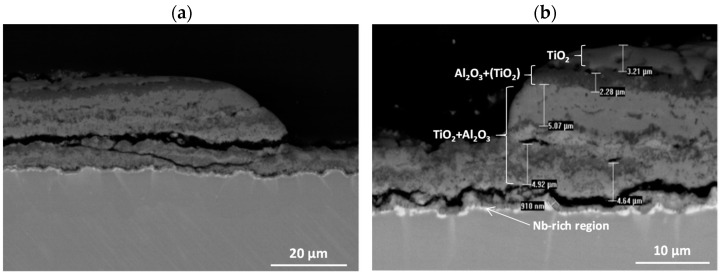
SEM-BSE images of Ti–46Al–7Nb–0.7Cr–0.1Si–0.2Ni alloy after cyclic oxidation at 975 °C (surface prepared by grinding): (**a**) low-magnification, (**b**) high-magnification.

**Figure 20 materials-15-02137-f020:**
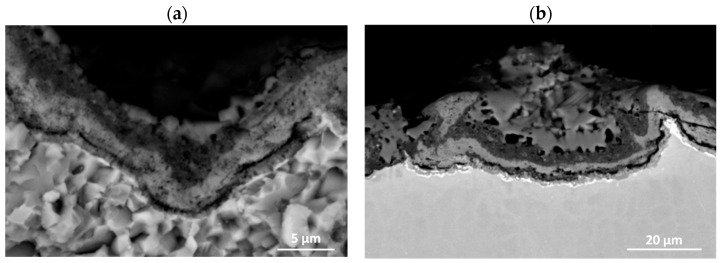
SEM-BSE images of Ti–46Al–7Nb–0.7Cr–0.1Si–0.2Ni alloy after cyclic oxidation at 975 °C (surface prepared by abrasive blasting): (**a**) high-magnification, (**b**) low-magnification.

**Figure 21 materials-15-02137-f021:**
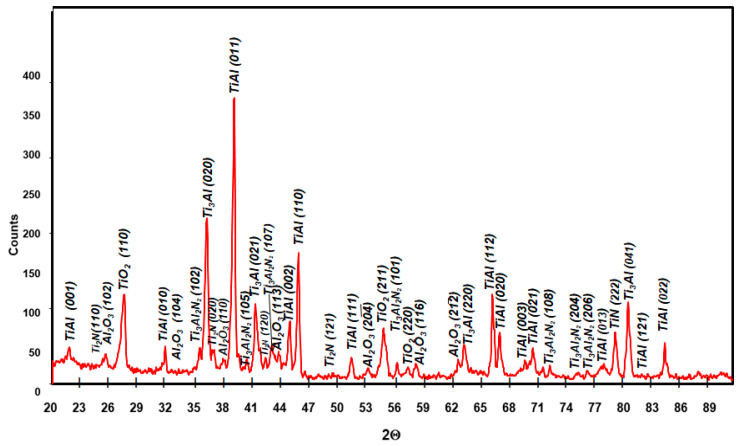
XRD pattern showing the predominant phases formed during oxidation of Ti–46Al–7Nb–0.7Cr–0.1Si–0.2Ni alloy at 875 °C (scale was removed from the samples).

**Figure 22 materials-15-02137-f022:**
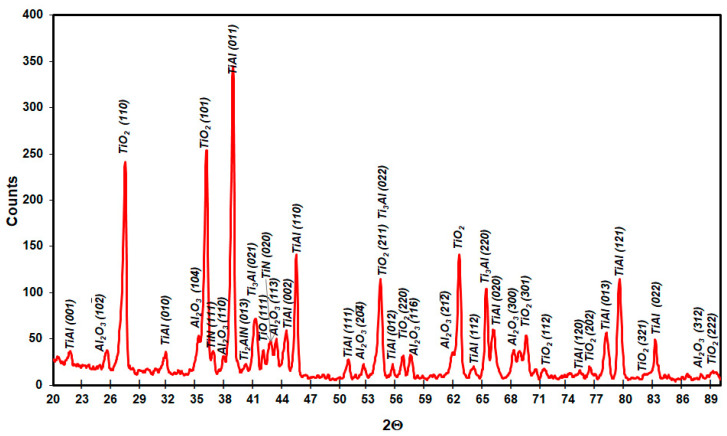
XRD pattern showing the predominant phases formed during oxidation of Ti–46Al–7Nb–0.7Cr–0.1Si–0.2Ni alloy at 900 °C.

**Figure 23 materials-15-02137-f023:**
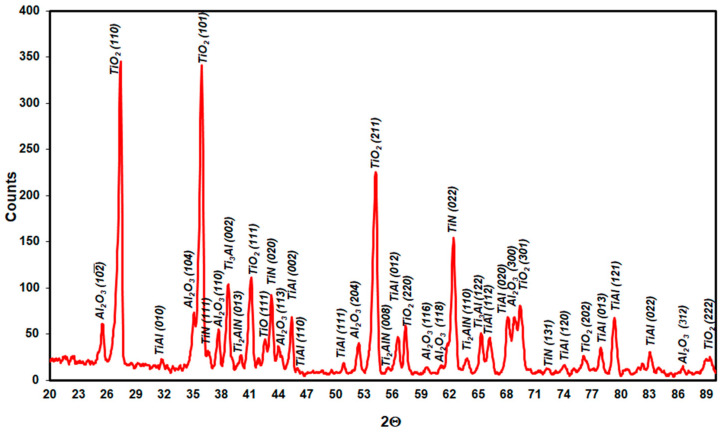
XRD pattern showing the predominant phases formed during oxidation of Ti–46Al–7Nb–0.7Cr–0.1Si–0.2Ni alloy at 950 °C.

**Figure 24 materials-15-02137-f024:**
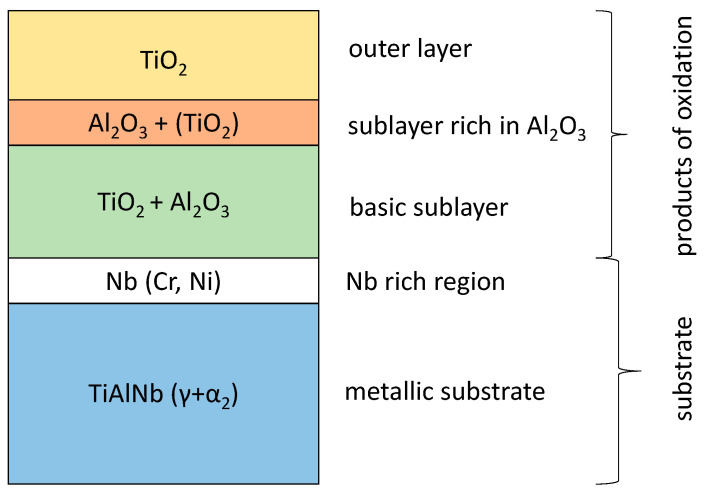
General structure of scale after oxidation of Ti-46Al-7Nb-0.7Cr-0.1Si-0.2Ni alloy.

**Figure 25 materials-15-02137-f025:**
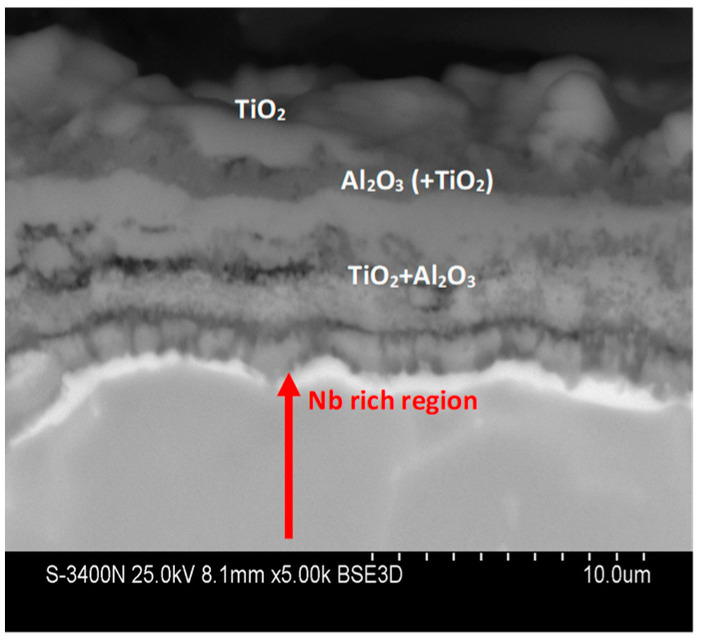
SEM-BSE images of Ti–46Al–7Nb–0.7Cr–0.1Si–0.2Ni alloy after cyclic oxidation.

**Table 1 materials-15-02137-t001:** EDS analysis in points marked in Figure 8 (mean values and standard deviations).

	Element	Ti	Al	Nb	Cr	Ni	Si
Chemical composition from the location #1	%mass	75.9 ± 2.16	23.1 ± 0.67	-	-	-	1.0 ± 0.12
	%at.	64.0 ± 1.69	34.6 ± 0.85	-	-	-	1.4 ± 0.16
Chemical composition from the location #2	%mass	43.3 ± 1.01	50.0 ± 1.13	6.7 ± 0.7	-	-	-
	%at.	32.0 ± 0.92	65.5 ± 2.03	2.5 ± 0.28	-	-	-
Chemical composition from the location #3	%mass	73.1 ± 2.61	17.1 ± 0.53	9.8 ± 0.62	-	-	-
	%at.	67.4 ± 1.72	27.0 ± 0.84	5.6 ± 0.76	-	-	-
Chemical composition from the location #4	%mass	44.5 ± 1.12	25.2 ± 0.64	26.1 ± 0.69	1.3 ± 0.13	2.9 ± 0.33	-
	%at.	42.9 ± 0.97	41.5 ± 0.86	12.7 ± 0.42	0.6 ± 0.04	2.3 ± 0.26	-

**Table 2 materials-15-02137-t002:** EDS analysis results in points marked in Figure 12 (mean values and standard deviations).

	Element	Ti	Al	Nb	Cr	Ni
Chemical composition from the location #A	%mass	85.4 ± 3.01	14.6 ± 0.43	-	-	-
	%at.	76.7 ± 2.18	23.3 ± 0.71	-	-	-
Chemical composition from the location #B	%mass	61.7 ± 1.72	10.5 ± 0.73	27.8 ± 0.91	-	-
	%at.	65.1 ± 1.79	19.7 ± 0.43	15.2 ± 0.42	-	-
Chemical composition from the location #C	%mass	61.5 ± 1.45	22.3 ± 0.58	14.0 ± 0.40	0.9 ± 0.05	1.3 ± 0.15
	%at.	55.8 ± 1.12	35.9 ± 1.01	6.5 ± 0.06	0.8 ± 0.48	1.0 ± 0.13

**Table 3 materials-15-02137-t003:** EDS analysis results in points marked in Figure 15 (mean values and standard deviations).

	Element	Ti	Al	Nb	Cr	Ni
Chemical composition from the location #1	%mass	92.8 ± 2.96	7.2 ± 0.48	-	-	-
	%at.	87.9 ± 3.01	12.1 ± 1.01	-	-	-
Chemical composition from the location #2	%mass	39.1 ± 1.21	55.2 ± 1.23	5.7 ± 0.7	-	-
	%at.	27.9 ± 1.03	70.0 ± 1.76	2.1 ± 0.27	-	-
Chemical composition from the location #3	%mass	61.3 ± 1.71	24.9 ± 0.74	13.8 ± 0.56	-	-
	%at.	54.4 ± 1.56	39.3 ± 0.83	6.3 ± 0.23	-	-
Chemical composition from the location #4	%mass	29.6 ± 0.98	27.4 ± 0.85	37.2 ± 0.81	2.3 ± 0.28	3.5 ± 0.33
	%at.	28.9 ± 0.99	47.5 ± 1.13	18.7 ± 0.53	2.1 ± 0.29	2.8 ± 0.21
Chemical composition from the location #5	%mass	49.3 ± 1.15	32.3 ± 0.89	17.8 ± 0.54	0.6 ± 0.05	-
	%at.	42.3 ± 0.98	49.3 ± 1.02	7.9 ± 0.61	0.5 ± 0.02	-
Chemical composition from the location #6	%mas	45.6 ± 1.03	33.2 ± 0.76	20.3 ± 0.44	0.9 ± 0.05	-
	%at.	39.4 ± 0.78	50.8 ± 1.02	9.0 ± 0.32	0.7 ± 0.03	-

## Data Availability

The data presented in this study are available on request from the corresponding author.

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
