# Peer review of "Resistance to High-Temperature Oxidation of Ti-Al-Nb Alloys"

_materials, 2022, doi:10.3390/ma15062137_

Round 1
Reviewer 1 Report
The paper "Resistance to high-temperature oxidation of alloys Ti-Al-Nb alloys based on intermetallic phases" it is suitable for publication in Materials Journal. It can be accepted in present form, because the author presented sufficient discussions regarding the microstructure of the oxidized layers and the influence of the Nb. If the author can add ICDD files for the XRD, should be great.
Good work
Author Response
I would like to thank the reviewer for your time and help. First, I really appreciate you for your comments about my paper. I added Miller indices for each phase (Fig. 21-23 in new manuscript)

Reviewer 2 Report
The manuscript entitled: Resistance to high-temperature oxidation of alloys Ti-Al-Nb alloys based on intermetallic phases considers the possibility of improving the oxidation resistance by the addition of Nb and the sensitivity of these alloys to cyclic oxidation. My concerns with the manuscript are as follows:
- How are the samples produced? It is not explained clearly in the Materials and test methods part.
- XRD pattern and SEM images of the original parent material should be introduced.
- Scale bars in Figs. 5-7, 9-12, 14, 16-18 are not clearly visible.
- Y-axis legend is missing in Figs. 8, 13, and 15.
- Table 2 - % mas should be written as %mass and %at should be written as %at. Applies to Tables 1, and 3 as well.
- Fig. 20-22 - X-axis legends units are missing and Y-axis legend with the unit is missing. The indexing of the peaks is not readable.
- There is no scientific explanation correlating the observed results and also not comparing with the published literature in an adequate fashion.
- Referencing needs an update. Recent articles should be introduced.
Author Response
I would like to thank the reviewer for your time and help. First, I really appreciate you for your comments about my paper. The following main changes are made based on the reviewer comments received:
- How are the samples produced? It is not explained clearly in the Materials and test methods part.
The material in the form of a cylinder with a diameter of about 69 mm was purchased from Flowserve Corporation Titanium and Reactive Metals Foundry (USA). The surfaces were abraded on abrasive paper of 800-grade or abrasive blasting using alundum of F40 grain and finally cleaned with acetone
(lines 87-93 in the new manuscript)
- XRD pattern and SEM images of the original parent material should be introduced.
The structure and XRD microanalysis result of Ti–46Al–7Nb–0.7Cr–0.1Si–0.2Ni is visible in Fig. 1.
- Scale bars in Figs. 5-7, 9-12, 14, 16-18 are not clearly visible.
The remark was applied according to the suggestions (Fig. 6a, 7-8, 10-13, 15, 17-20 in new manuscript)
- Y-axis legend is missing in Figs. 8, 13, and 15.
The remark was applied according to the suggestions (Fig. 9,14, 16 in new manuscript)
- Table 2 - % mas should be written as %mass and %at should be written as %at. Applies to Tables 1, and 3 as well.
The remark was applied according to the suggestions (Tab. 1-3)
- 20-22 - X-axis legends units are missing and Y-axis legend with the unit is missing. The indexing of the peaks is not readable.
The remark was applied according to the suggestions and Miller indices was added for each phase (Fig. 21-23 in new manuscript)
- There is no scientific explanation correlating the observed results and also not comparing with the published literature in an adequate fashion.
In general, oxidation is regulated by the inward diffusion of oxygen and nitrogen and the outward diffusion of Ti and Al. It is generally believed that small amounts of Nb, significantly reduce the kinetics of oxidation in air, favoring the formation of a continuous layer of nitride at the oxide / metal interface that controls downstream oxidation.
For example, similar results were observed during the oxidation of the Ti-47Al-2Cr-2Nb and Ti-45Al-2Cr-8Nb alloys [22] and Ti-48Al-2Cr-2Nb alloy [23]. There are some papers devoted the influence of Nb on the oxidation behaviour, so two papers was added to references.
- Referencing needs an update. Recent articles should be introduced.
In general, references were selected taking into account two main criteria, i.e. the same materials and comparable oxidation time, and in the manuscript are five recent articles:
[3] published in 2021,
[4] published in 2021,
[6] published in 2018,
[8] published in 2018,
[11] published in 2022.
Reviewer 3 Report
The manuscript deals with the resistance of the Ti-46Al-7Nb-0.7Cr-0.1Si-0.2Ni alloy to high temperature oxidation. Although experimental results are extensive and well discussed, the manuscript needs to be refined and can be considered to be published after the mandatory major revision. The main comments are given below:
- The title should be corrected – the word “alloys” is used for two times. I would also avoid the part “based on the intermetallic phases”.
- Line 20: please include the specific applications.
- Line 22, better: “…largely concern the improving…”.
- It would be appropriate to write the atomic content as follows: at.% or at. %. Please use this denotation through the whole manuscript.
- Lines 60-72 should be moved to the “Introduction” section in the appropriate text form.
- Line 70, please correct: “…, the mechanism of corrosion at high temperature…”.
- In the “Materials and Methods” section, details related to SEM/EDS, XRD (anode, parameters, etc.) are missing and should be added.
- What is the initial state of samples (as-cast, as-annealed)? Which method was used to prepare the sample (arc melting, induction melting, etc.)? How did you ensure the homogeneity of the sample? This information should be included in the “Materials and Methods” section.
- What kind of furnace did you use for samples oxidation? How did you measure the mass change per unit area? The proper description of the measurement is missing and should be also included in the “Materials and Methods” section.
- Line 77: The sentence “Cyclic oxidation experiments were realized using temperatures range of 875 °C - 975 °C for 60 cycles in atmospheric air.” is a little bit confusing. Please rewrite the sentence, or include specific temperature values in parentheses.
- Line 78, better: “…annealing at this temperature for 1 hour…”.
- Line 82: “…Ra, value…”.
- Line 83, better: “The microstructure…”.
- Line 84: “…observed using a…” instead of “…observed in a…”.
- I wonder why the author used Ref. [22] corresponding to unpublished results instead of adding the author of Ref. [22] in the authorship. Please, clarify.
- The element content in the Ti-48Al-2Cr-2Nb reference alloy is different from the investigated alloy. Thus, the content of Cr 0.7 vs 2 could also possibly affect the oxidation behavior. Therefore, the comparison is not entirely appropriate. Moreover, the reference alloy was investigated at different temperature range, thus temperatures below 875 °C are not relevant in the comparison. They are rather informative. Please, give the statement.
- The tick markers should be added to axes in Figs. 1-4.
- It is better use the term “cycles” in the text instead of “hours”, e.g. line 102. Although the annealing time was 1 hour during every cycle, the oxidation appears also during heating and cooling of the sample. Therefore, the actual time of oxidation is higher than 60 hours (relevant for line 102).
- Lines 102-103: “The products formed of the Ti-46Al-7Nb-0.7Cr-0.1Si-0.2Ni alloy show good adhesion to the substrate throughout the test at these temperatures”. What is the evidence of this premise?
- Line 116: “…decreases…”.
- In the description of Fig. 4 you state that “The higher surface roughness (Ra=5.8 µm) contributes to a slight increase in mass gain.”. However, the trend in Fig. 4 is not so obvious and is valid for the 900 °C experiment only. At first cycles of 975 °C experiment, finer roughness has higher mass gain. At 925 °C, the finer roughness has rather higher mass gain throughout the cycles than 5.8 µm. Please, give the statement to this. Did you exclude the possible measurement error by repeating the experiment at particular temperatures (at least for one sample)?
- In the chapter 3.2 it is not clear, whether you show the results of samples with initial roughness of 0.06 or 5.8 µm. Consider to designate the samples to be easily distinguished.
- In the figure caption of Fig. 5 you mention TiO2. However, no proof is shown. How did you identify crystallites? The EDS method is not sufficient. Based on chemical composition measured by EDS, it can be only assumed which structure occurs. If you identified the crystallites using X-ray diffraction, it should me mentioned in the text, or you should refer to XRD results given at the end of chapter 4.
- You should avoid the term “X-ray microanalysis” (e.g. in line 154), since it can be confusing as you used various methods based on X-rays (XRD, EDS). Moreover, if the term “X-ray microanalysis” is meant to be the EDS method, you cannot be sure with this statement unless you confirm the structure by any of diffraction methods.
- Images with EDS spectra (e.g. Fig. 8) have quite low resolution. Basically, they are not necessary to be in the manuscript. Tables with values of chemical composition obtained by EDS are sufficient.
- Lines 153-162: The description is a little bit confusing. It would be suitable to give the information about particular EDS points into the text to the corresponding layer. Alternatively, you could mark the individual layers in Fig. 7b.
- Table 1: How many measurements did you perform within each layer? One measurement per layer is not sufficient, since the layers are relatively thin. Therefore, the results could be influenced by neighboring layers, since you measured a small volume by point EDS analysis (e.g. relevant also for x6 in Fig. 14b). If you performed more than one measurement per layer, please include the standard deviation values in corresponding tables in the whole manuscript.
- You describe the initial alloy as two-phase, however, you did not analyze the initial state of the alloy before oxidation. A reader should have the information about the microstructure, phase constitution and measured chemical alloy / phase composition of the initial alloy. In that case you could also discuss the possible relation of Fig. 10 to the initial microstructure, since the spallation could be possibly related to the individual phase regions.
- It would be better to move the Table 3 below Fig. 14 or 15.
- In Fig. 18b, you could highlight the individual layers confirmed by XRD and concluded from EDS.
- Scanned images (BSE) – better: SEM-BSE images, or BSE-SEM images
- XRD patterns:
- please include Miller indices of planes instead of d the spacing for each phase,
- the XRD pattern of initial state of the alloy before oxidation would be illustrative for comparison,
- please unify the graphics of XRD patterns. Tick markers for 2 Theta axis should be added. Horizontal lines as well as values of Intensity including tick markers could be removed and replaced by a.u. The XRD images could also be sorted by the annealing temperature in the manuscript.
- Please correct the number of the “Summary” chapter to 4.
- Line 299: “In order to determine the range of resistance to oxidation in air of Ti-46Al-7Nb-0.7Cr-299 0.1Si-0.2Ni alloy…” – better: “In order to determine the range of resistance of the Ti-46Al-7Nb-0.7Cr-299 0.1Si-0.2Ni alloy to oxidation in air…”.
- Lines 355-356: The sentence is not clear, please rewrite it.
- Line 360: The EDS maps of the cross section would gain the info about the diffusion of oxygen and nitrogen into the substrate.
- Although the English of the text is generally good, a native English speaker should go over the text.
Author Response
I would like to thank the reviewer for your time and help. First, I really appreciate you for your comments about my paper. The following main changes are made based on the reviewer comments received:
- The title should be corrected – the word “alloys” is used for two times. I would also avoid the part “based on the intermetallic phases”.
The title was corrected as requested.
- Line 20: please include the specific applications
The remark was applied according to the suggestions (lines 20-21 in new manuscript)
- Line 22, better: “…largely concern the improving…”.
It was done as requested (line 24 in new manuscript )
- It would be appropriate to write the atomic content as follows: at.% or at. %. Please use this denotation through the whole manuscript.
It was corrected and used: “%at.”
- Lines 60-72 should be moved to the “Introduction” section in the appropriate text form.
It was done as requested (lines 44-54 in new manuscript)
- Line 70, please correct: “…, the mechanism of corrosion at high temperature…”.
It was done as requested (lines 82-32 in new manuscript)
- In the “Materials and Methods” section, details related to SEM/EDS, XRD (anode, parameters, etc.) are missing and should be added.
It was done as requested (lines 114-118 in new manuscript)
- What is the initial state of samples (as-cast, as-annealed)? Which method was used to prepare the sample (arc melting, induction melting, etc.)? How did you ensure the homogeneity of the sample? This information should be included in the “Materials and Methods” section.
This alloy was prepared by an arc melting process. In order to remove the internal defects and obtain a homogenised of the sample, the castings was processed by the HIP (Hot Isostatic Pressing) method.
(Lines 87-91 in new manuscript)
- What kind of furnace did you use for samples oxidation? How did you measure the mass change per unit area? The proper description of the measurement is missing and should be also included in the “Materials and Methods” section.
The samples were heated up in the desired atmosphere with the tube furnace (high accuracy temperature sensors: ±1°C), then annealed in a given temperature, and subsequently cooled down to the room temperature. The gravimetric method was selected as a method of measuring the reaction speed, control of weight change was performed on an RADWAG precision scale with an accuracy 10-4 g. Trials were repeated three times and the presented test results are averaged.
(Lines 100-103 and 106-109)
- Line 77: The sentence “Cyclic oxidation experiments were realized using temperatures range of 875 °C - 975 °C for 60 cycles in atmospheric air.” is a little bit confusing. Please rewrite the sentence, or include specific temperature values in parentheses.
It was done as requested (lines 99-100)
- Line 78, better: “…annealing at this temperature for 1 hour…”.
It was done as requested (line 104 in new manuscript)
- Line 82: “…Ra, value…”.
It was done as requested (line 112 in new manuscript)
- Line 83, better: “The microstructure…”.
It was done as requested (line 113 in new manuscript)
- Line 84: “…observed using a…” instead of “…observed in a…”.
It was done as requested (line 113 in new manuscript)
- I wonder why the author used Ref. [22] corresponding to unpublished results instead of adding the author of Ref. [22] in the authorship. Please, clarify.
Author of Ref [24 in new manuscript] is dead and the obtained results wasn't published.
- The element content in the Ti-48Al-2Cr-2Nb reference alloy is different from the investigated alloy. Thus, the content of Cr 0.7 vs 2 could also possibly affect the oxidation behavior. Therefore, the comparison is not entirely appropriate. Moreover, the reference alloy was investigated at different temperature range, thus temperatures below 875 °C are not relevant in the comparison. They are rather informative. Please, give the statement.
It is well known that small amounts of Nb significantly reduce the kinetics of oxidation in air, favoring the formation of a continuous layer of nitride at the oxide / metal interface that controls downstream oxidation and similar results were observed during the oxidation of the Ti-47Al-2Cr-2Nb and Ti-45Al-2Cr-8Nb alloys [22]. Niobium seems to be especially important because it concentrates in the base near the phase boundary and limits formation of the solid solution α-Ti (Al,N,O). Niobium being a β forming elements in relation to titanium promotes formation of a low soluble phase β -Ti(Al,N,O) of nitrogen and oxygen.
It limits diffusion of interstitial elements to the core, and as a consequence, it slows down the increase of oxidation products.
This paper presents the results of reference alloy at temperature below 875 C only for informative results, not for in comparison.
- The tick markers should be added to axes in Figs. 1-4.
It was done as requested (Fig. 2-5 in new manuscript)
- It is better use the term “cycles” in the text instead of “hours”, e.g. line 102. Although the annealing time was 1 hour during every cycle, the oxidation appears also during heating and cooling of the sample. Therefore, the actual time of oxidation is higher than 60 hours (relevant for line 102).
It was done as requested (line 138 in new manuscript)
- Lines 102-103: “The products formed of the Ti-46Al-7Nb-0.7Cr-0.1Si-0.2Ni alloy show good adhesion to the substrate throughout the test at these temperatures”. What is the evidence of this premise?
Oxidation in 900°C and 925°C, results only in mass increment. For 975°C, mass increment appears only during initial cycles and then mass decrement dominates caused by chipping in the cooling cycle.
- Line 116: “…decreases…”.
It was done as requested (line 153 in new manuscript)
- In the description of Fig. 4 you state that “The higher surface roughness (Ra=5.8 µm) contributes to a slight increase in mass gain.”. However, the trend in Fig. 4 is not so obvious and is valid for the 900 °C experiment only. At first cycles of 975 °C experiment, finer roughness has higher mass gain. At 925 °C, the finer roughness has rather higher mass gain throughout the cycles than 5.8 µm. Please, give the statement to this. Did you exclude the possible measurement error by repeating the experiment at particular temperatures (at least for one sample)?
The remark was applied according to the suggestions (line 169 in new manuscript).
In all tests at least three samples were used and each oxidation test was repeated at least three times.
- In the chapter 3.2 it is not clear, whether you show the results of samples with initial roughness of 0.06 or 5.8 µm. Consider to designate the samples to be easily distinguished.
It was done as requested (new signatures: Fig. 6a, 7, 8, 10-12, 15, 17-20)
- In the figure caption of Fig. 5 you mention TiO2. However, no proof is shown. How did you identify crystallites? The EDS method is not sufficient. Based on chemical composition measured by EDS, it can be only assumed which structure occurs. If you identified the crystallites using X-ray diffraction, it should me mentioned in the text, or you should refer to XRD results given at the end of chapter 4.
The shape of these crystallites is identical to those observed on titanium and identified as an allotropic variety of TiO2 called rutile. The results confirm the presence of TiO2 rutile (new Fig - Fig. 6b in new manuscript).
- You should avoid the term “X-ray microanalysis” (e.g. in line 154), since it can be confusing as you used various methods based on X-rays (XRD, EDS). Moreover, if the term “X-ray microanalysis” is meant to be the EDS method, you cannot be sure with this statement unless you confirm the structure by any of diffraction methods.
It was done as requested (line 200 in new manuscript)
- Images with EDS spectra (e.g. Fig. 8) have quite low resolution. Basically, they are not necessary to be in the manuscript. Tables with values of chemical composition obtained by EDS are sufficient.
According to the suggest of another reviewier the Y-axis legend was added in EDS.
- Lines 153-162: The description is a little bit confusing. It would be suitable to give the information about particular EDS points into the text to the corresponding layer. Alternatively, you could mark the individual layers in Fig. 7b.
The remark was applied according to the suggestions (Fig. 8b in the new manuscript)
- Table 1: How many measurements did you perform within each layer? One measurement per layer is not sufficient, since the layers are relatively thin. Therefore, the results could be influenced by neighboring layers, since you measured a small volume by point EDS analysis (e.g. relevant also for x6 in Fig. 14b). If you performed more than one measurement per layer, please include the standard deviation values in corresponding tables in the whole manuscript.
Trials were repeated three times. Results are presented as the mean and was added standard deviation (Tab 1-3 in new manuscript)
- You describe the initial alloy as two-phase, however, you did not analyze the initial state of the alloy before oxidation. A reader should have the information about the microstructure, phase constitution and measured chemical alloy / phase composition of the initial alloy. In that case you could also discuss the possible relation of Fig. 10 to the initial microstructure, since the spallation could be possibly related to the individual phase regions.
The structure and XRD microanalysis result of Ti–46Al–7Nb–0.7Cr–0.1Si–0.2Ni is visible in Fig. 1 in new manuscript
- It would be better to move the Table 3 below Fig. 14 or 15.
It was done as requested
- In Fig. 18b, you could highlight the individual layers confirmed by XRD and concluded from EDS.
The remark was applied according to the suggestions (Fig. 19b in the new manuscript)
- Scanned images (BSE) – better: SEM-BSE images, or BSE-SEM images
It was done as requested (Fig. 8, 15, 17, 19, 20 in new manuscript)
- XRD patterns:
- please include Miller indices of planes instead of d the spacing for each phase,
- the XRD pattern of initial state of the alloy before oxidation would be illustrative for comparison,
- please unify the graphics of XRD patterns. Tick markers for 2 Theta axis should be added. Horizontal lines as well as values of Intensity including tick markers could be removed and replaced by a.u. The XRD images could also be sorted by the annealing temperature in the manuscript.
It was done as requested (Fig. 1 and 21-23 in new manuscript)
- Please correct the number of the “Summary” chapter to 4.
It was done as requested
- Line 299: “In order to determine the range of resistance to oxidation in air of Ti-46Al-7Nb-0.7Cr-299 0.1Si-0.2Ni alloy…” – better: “In order to determine the range of resistance of the Ti-46Al-7Nb-0.7Cr-299 0.1Si-0.2Ni alloy to oxidation in air…”.
It was done as requested (lines 365-366 in new manuscript)
- Lines 355-356: The sentence is not clear, please rewrite it.
This sentence was corrected as:
Rutile has a defective anionic sublattice and there the oxygen in-core diffusion takes place. Oxygen diffuses toward the product-metallic substrate interface, causing an increase in the thickness of the oxide layer due to the formation of titanium and aluminum oxides in this layer. The occurrence of defects in the rutile depends on the partial pressure of oxygen. At higher temperatures, it is the in-core diffusion of oxygen (through the doubly and singly ionized oxygen vacancies of the rutile) and the out-core diffusion of the titanium, aluminum, and alloying elements that occurs.
(lines 416-423 in new manuscript)
- Line 360: The EDS maps of the cross section would gain the info about the diffusion of oxygen and nitrogen into the substrate.
In these research didn't make EDS maps of the cross section. I will extend these investigations in next research.
- Although the English of the text is generally good, a native English speaker should go over the text.
The corrections were applied according to the another reviewier report.
Reviewer 4 Report
The paper introduces an experimental investigation on the effect of niobium addition kinetics as a high-melting element causing improved oxidation resistance, contributing to the reduction of the oxidation rate. In my opinion the paper needs a several improvements before to be reconsidered for possible publication:
- Material and test methods.
- When the authors write “Each traction cycle consisted of heating the samples to a specific temperature” is necessary which instrument is used.
- The authors should report the uncertainties of the instruments.
General comments:
- I have attached the .pdf file to report the major corrections about the English form. However, the authors should improve the English form of the paper.

Author Response
I would like to thank the reviewer for your time and help. First, I really appreciate you for your comments about my paper. The following main changes are made based on the reviewer comments received:
- Material and test methods.
- When the authors write “Each traction cycle consisted of heating the samples to a specific temperature” is necessary which instrument is used.
It was done as requested (lines 101 in new manuscript)
- The authors should report the uncertainties of the instruments.
It was done as requested (line 102 in new manuscript)
General comments:
- I have attached the .pdf file to report the major corrections about the English form. However, the authors should improve the English form of the paper.
The remarks were applied according to the suggestions
Round 2
Reviewer 3 Report
The manuscript has been significantly improved, however, the major revision is still mandatory. The main comments are given below:
- Fig. 1 should be included in the “Results” section. The addition of separate dedicated section (within Chapter 3) placed before oxidation tests is welcomed. Write a paragraph with a description / explanation of results shown in Fig. 1. Phases identified in the corresponding XRD pattern should be also assigned to the particular microstructure constituents in the SEM image.
- XRD parameters in the “Materials and Methods” section are still missing. Please, specify the anode material, step size, time per step, etc.
- Some of the information from the answer to my 16th comment from the previous revision (please, see below) could be also included in the manuscript as the explanation why the comparison is informative only.
It is well known that small amounts of Nb significantly reduce the kinetics of oxidation in air, favoring the formation of a continuous layer of nitride at the oxide / metal interface that controls downstream oxidation and similar results were observed during the oxidation of the Ti-47Al-2Cr-2Nb and Ti-45Al-2Cr-8Nb alloys [22]. Niobium seems to be especially important because it concentrates in the base near the phase boundary and limits formation of the solid solution α-Ti (Al,N,O). Niobium being a β forming elements in relation to titanium promotes formation of a low soluble phase β -Ti(Al,N,O) of nitrogen and oxygen.
It limits diffusion of interstitial elements to the core, and as a consequence, it slows down the increase of oxidation products.
This paper presents the results of reference alloy at temperature below 875 C only for informative results, not for in comparison.
- Standard deviations shown in tables with chemical composition measured by EDS should be given to the each mean value (i.e. separately for Ti, Al, etc.). It can be assumed that standard deviations will not show the same values within elements in one line. It is also necessary to distinguish standard deviations for measurements in mass % as well as atomic %. Thus, the form of standard deviation should look like (e.g. for the first line in Table 1):
Ti Al …
%mass 75.9 ± XX.X 23.1 ± ZZ.Z ...
%at. 64.0 ± YY.Y 34.6 ± VV.V ...
Explanation: XX.X – value of standard deviation valid for mean value of Ti in mass % calculated from 3 measurements, YY.Y – the same for Ti in at. %, etc.
- Please, improve the aesthetics of XRD patterns images. Try to avoid the overlapping of peak assignment with peaks (Figs. 6b, 21-23) or axes of the pattern (Fig. 6b). If necessary, you can use reference lines between the peak and phase assigned to that peak.
- Figure captions for XRD patterns, better: “XRD pattern showing the predominant…”
- Please, check the number of spaces in figure captions with added expression “SEM-BSE images”.
